# Phylodynamics of SARS-CoV-2 in France, Europe, and the world in 2020

Romain Coppée[1]*, François Blanquart[1,2], Aude Jary[3], Valentin Leducq[3], Valentine Marie Ferré[1,4], Anna Maria Franco Yusti[1,4], Léna Daniel[1,4], Charlotte Charpentier[1,4], Samuel Lebourgeois[1,4], Karen Zafilaza[3,5], Vincent Calvez[3,5], Diane Descamps[1,4], Anne-Geneviève Marcelin[3,5], Benoit Visseaux[1,4], Antoine Bridier-Nahmias[1]*

[1]Université Paris Cité and Sorbonne Paris Nord, Inserm, IAME, Paris, France; [2]Centre for Interdisciplinary Research in Biology (CIRB), Collège de France, CNRS Inserm, PSL Research University, Paris, France; [3]Sorbonne Université, Inserm, iPLESP, Paris, France; [4]Service de Virologie, Hôpital Bichat Claude Bernard, AP-HP, Paris, France; [5]Service de Virologie, Hôpital de la Pitié-Salpêtrière, AP-HP, Paris, France

**\*For correspondence:**
romain.coppee@inserm.fr (RC);
antoine.bridier-nahmias@inserm.fr (AB-N)

**Competing interest:** The authors declare that no competing interests exist.

**Abstract** Although France was one of the most affected European countries by the COVID-19 pandemic in 2020, the dynamics of severe acute respiratory syndrome coronavirus 2 (SARS-CoV-2) movement within France, but also involving France in Europe and in the world, remain only partially characterized in this timeframe. Here, we analyzed GISAID deposited sequences from January 1 to December 31, 2020 (*n* = 638,706 sequences at the time of writing). To tackle the challenging number of sequences without the bias of analyzing a single subsample of sequences, we produced 100 subsamples of sequences and related phylogenetic trees from the whole dataset for different geographic scales (worldwide, European countries, and French administrative regions) and time periods (from January 1 to July 25, 2020, and from July 26 to December 31, 2020). We applied a maximum likelihood discrete trait phylogeographic method to date exchange events (i.e., a transition from one location to another one), to estimate the geographic spread of SARS-CoV-2 transmissions and lineages into, from and within France, Europe, and the world. The results unraveled two different patterns of exchange events between the first and second half of 2020. Throughout the year, Europe was systematically associated with most of the intercontinental exchanges. SARS-CoV-2 was mainly introduced into France from North America and Europe (mostly by Italy, Spain, the United Kingdom, Belgium, and Germany) during the first European epidemic wave. During the second wave, exchange events were limited to neighboring countries without strong intercontinental movement, but Russia widely exported the virus into Europe during the summer of 2020. France mostly exported B.1 and B.1.160 lineages, respectively, during the first and second European epidemic waves. At the level of French administrative regions, the Paris area was the main exporter during the first wave. But, for the second epidemic wave, it equally contributed to virus spread with Lyon area, the second most populated urban area after Paris in France. The main circulating lineages were similarly distributed among the French regions. To conclude, by enabling the inclusion of tens of thousands of viral sequences, this original phylodynamic method enabled us to robustly describe SARS-CoV-2 geographic spread through France, Europe, and worldwide in 2020.

## Editor's evaluation

This paper is a comprehensive, quantitative, and robust overview of the global, European, and French genomic epidemiology of SARS-CoV-2 in the first year of the pandemic. It contributes methodological advances in maximum likelihood phylogeography, using multiple scales and providing a simulation-based validation. The results show two distinct patterns of SARS-CoV-2 exchange events

between the first and second half of 2020, with Europe being involved in most intercontinental exchanges: France experienced viral introductions primarily from North America and Europe during the first wave, while the second wave saw limited intercontinental movement and a significant contribution of the virus from Russia into Europe.

## Introduction

On December 1, 2019, an outbreak of severe respiratory disease was identified in the city of Wuhan, China (*Huang et al., 2020*). The severe acute respiratory syndrome coronavirus 2 (SARS-CoV-2) was rapidly identified as the agent of the disease (*Zhu et al., 2020*), responsible for the ongoing global pandemic of coronavirus disease 2019 (COVID-19). By the end of 2020, the virus caused over 1.8 million deaths worldwide including ~65,000 deaths in France, concomitantly with social and economic devastations in many regions of the world (*Mofijur et al., 2021*; *Santomauro et al., 2021*). Since the beginning of COVID-19 pandemic, the scientific community has thoroughly characterized the virus, including its pathogenesis, the monitoring of its circulation in human populations, and the development of several treatments or vaccines (*Cevik et al., 2020*; *Krammer, 2020*). Epidemiological models have been particularly helpful to quantify viral spread both in the short and long terms and to inform public health decisions (*Hoertel et al., 2020*; *Kissler et al., 2020*).

In addition to clinical and epidemiological insights, viral whole-genome sequencing has become a powerful and invaluable tool to better understand infection dynamics (*Volz et al., 2013*), including the COVID-19 pandemic. The number of available SARS-CoV-2 whole-genome sequences has rapidly grown thanks to the efforts of scientists and researchers gathered via international networks such as the Global Initiative on Sharing All Influenza Data, GISAID (https://www.gisaid.org/; *Khare et al., 2021*). These genomic sequences are essential to effectively reconstruct the global viral spread and the origins of variants. Genomic data have become a strong asset in addition to epidemiological data to inform governments and help public health decisions (*Attwood et al., 2022*; *Rife et al., 2017*). However, due to the computational time required for many analyses, existing phylogenetic tools are limited for studying large amounts of data such as those generated by widespread viral sequencing. Therefore, it is still necessary to develop methods to analyze large datasets while optimizing computational calculation times. Producing appropriate subsamples through several replicates may be an efficient approach in this matter.

In France, the first COVID-19 suspected case was identified in late December 2019 (*Deslandes et al., 2020*), and the first confirmed cases of SARS-CoV-2 infection were detected on January 24, 2020, in individuals who had recently traveled in China (*Bernard Stoecklin et al., 2020*). COVID-19 cases remained scarce until the end of February, when the national incidence curve of new SARS-CoV-2 infections started to rise (*Figure 1*). By the end of February, reinforced measures were announced, including social distancing, cessation of passenger flights to France, school closure, and finally, a complete lockdown across the entire country from March 17 to May 10, 2020. The reported daily incidence and numbers of severe cases peaked at the beginning of April 2020 before decreasing steadily until August 2020. However, after the relaxation of social distancing measures in June, a second wave of infections occurred in early September peaking at more than 100,000 positive cases and 1300 confirmed deaths in a single day on November 2, 2020 (*Figure 1*). After this peak, daily incidence and severe COVID-19 cases gradually diminished down to a number of positive daily cases varying between 2000 and 25,000 at the end of 2020 thanks to a second national lockdown applied between October 29 and December 15, 2020. Epidemiological trends were similar in most European countries except for Russia or Romania, where high rates of SARS-CoV-2-related deaths were reported even in the summer of 2020. Of note, the other continents showed different patterns of virus circulation: compared to Europe, the number of deaths increased about 2 weeks later in North America and remained high throughout 2020; and from early May, Asia and South America were also highly impacted by the pandemic (*Figure 1—figure supplement 1*).

Elucidating the SARS-CoV-2 dynamic throughout the various phases of the pandemic is paramount to better anticipate how to limit virus circulation for future viral epidemics (*Rife et al., 2017*). Here, we analyzed GISAID deposited sequences to elucidate the origins and spread of the virus in France, Europe, and the world from January 1 to December 31, 2020. Through a maximum likelihood discrete trait phylogeographic method, we estimated the main geographical areas that contributed to viral

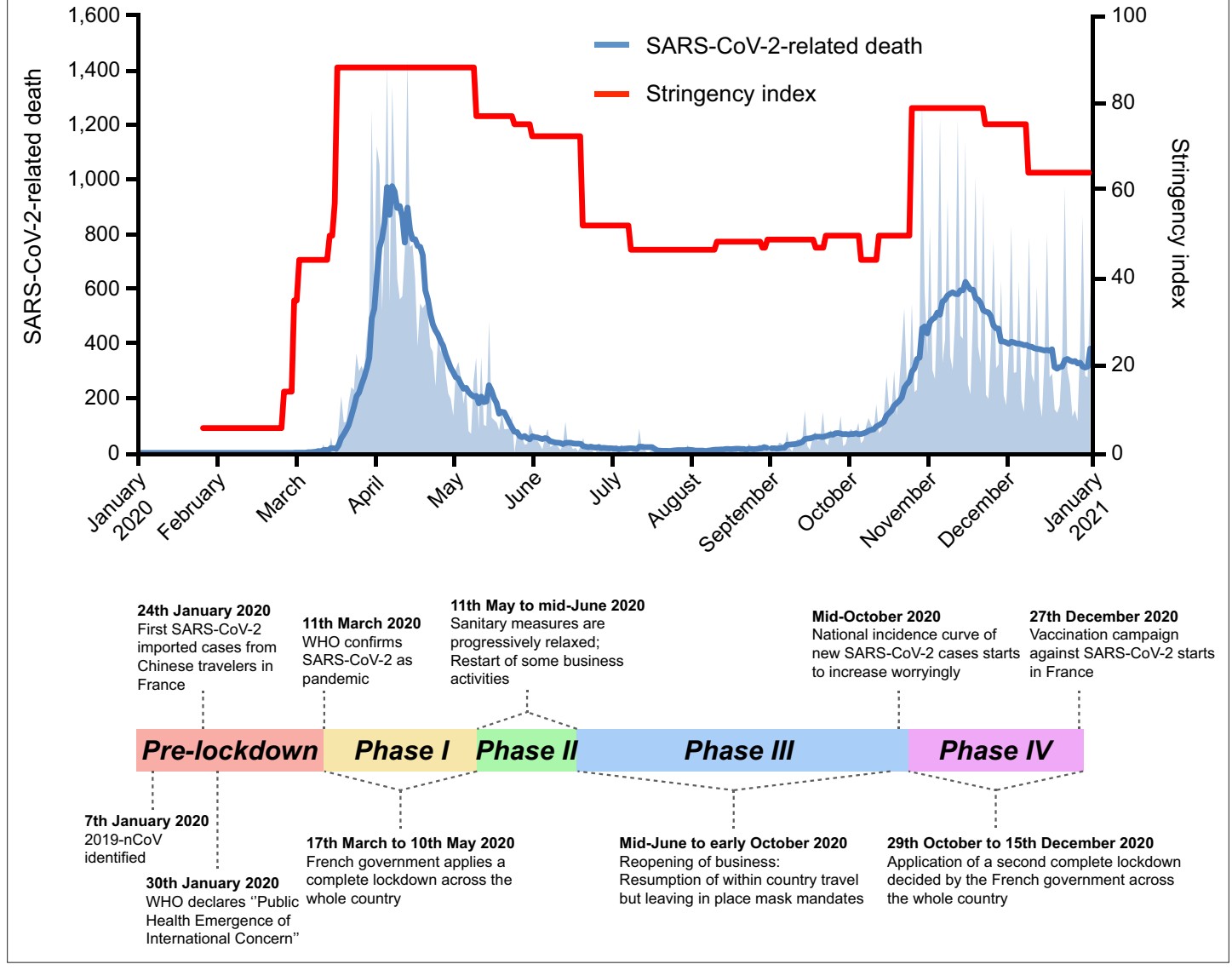

**Figure 1.** Timeline of severe acute respiratory syndrome coronavirus 2 (SARS-CoV-2)-related deaths and stringency index in France, 2020. Key events are indicated on the timeline. Official lockdowns included stay home orders and closure of schools and daycares. Based on SARS-CoV-2-related deaths, the two first French epidemic waves are, respectively, dated from March to July 2020, and September to December 2020. SARS-CoV-2-related deaths are displayed as the daily number of deaths (light blue area) and as the weekly average of daily number of deaths (dark blue curve). The stringency (Oxford) index is a composite measure based on different response indicators including school and workplace closures and travel bans, rescaled to a value from 0 to 100 (100 = strictest) (*Hale et al., 2021*).

The online version of this article includes the following figure supplement(s) for figure 1:

**Figure supplement 1.** Severe acute respiratory syndrome coronavirus 2 (SARS-CoV-2)-related deaths per week of 2020 for each (**A**) continent and (**B**) European country.

**Figure supplement 2.** Difference in the count of severe acute respiratory syndrome coronavirus 2 (SARS-CoV-2)-positive cases and SARS-CoV-2-related deaths over time.

introduction into France and Europe, the countries/continents to which France exported SARS-CoV-2 the most and the contribution of the different French regions to the national circulation of the virus. The main exchanged lineages were also investigated. We looked at the differences in virus circulation during each of the two European epidemic waves of 2020 independently. Given France's central geographic location in Europe and the high proportion of international travelers visiting this country before the pandemic, we aimed to explore the role that France played in SARS-CoV-2 exchanges both in Europe and worldwide.

## Results

### Defining appropriate subsamples using simulations

From January 1 to December 31, 2020, a total of 638,706 sequences were retained in our study. Inferring a phylogenic tree with such a large number of sequences would require very long calculation times. To overcome this limit, we constructed smaller datasets by randomly choosing subsamples (with replacement) of the sequences. The number of sequences for each country at each week was chosen to be proportional to the number of SARS-CoV-2-related deaths per country and per week with a 2-week shift to account for the time between infection and death.

As a proof of concept, we conducted an extensive simulation study to estimate the accuracy of the discrete trait phylogeographic inference for rates of transitions between two distinct locations. First, to evaluate the precision of such inference on a tree of 1000 leaves, we simulated a two-states model with different combinations of transition rates in 50 replicates. Parameters were correctly estimated with limited variability across the 50 replicates. The median parameters across replicates gave a very accurate estimation (*Figure 2A*).

Next, to evaluate how independent parameter estimates are done on randomly subsampled trees of the same larger phylogeny, we inferred parameters on 50 100-leaves trees randomly subsampled from a 10,000-leaves SARS-CoV-2 phylogenetic tree. For each resulting subtree, we conducted inferences on 50 replicates corresponding to 50 realizations of the stochastic process of evolution of the discrete character – as done in the first simulation – on the whole tree of 10,000 leaves. For each replicate, we observed some error on the estimation of the parameter, because one replicate only corresponds to one possible realization of the evolutionary process, although the overall median of inferred parameters across subsampled trees was closer to the true parameter values (*Figure 2B*).

Different estimations of the transition rates conducted on different subsampled trees are not expected to be fully independent because the subtrees partly share the same evolutionary history. Therefore, we estimated the level of independence of these estimations. When several estimates are perfectly independent from one another and are averaged to obtain the final estimate of the quantity of interest, we expect the error in parameter estimation to converge to 0 with a $1/N$ ($N^{-1}$) scaling, where $N$ is the number of replicates. This is indeed what we observed when we calculated the error on estimation of the parameter as a function of the chosen number of replicates $N$ in the first set of simulations. Here, the replicates were truly independent replicate realizations of the evolutionary history and inference was conducted on the whole tree of 1000 leaves (*Figure 2C*). On the contrary, when estimates are perfectly dependent, error on the averaged parameter estimate is expected to not decrease with $N$. When evaluating the error on parameter estimates across subsamples of the large tree, we expected the scaling of error as a function of number of subsamples $N$ to be intermediate between non-independence ($\sim N^0$ scaling) and perfect independence ($\sim N^{-1}$ scaling). Using the relationship between log(error) as a function of log($N$), we estimated a slope of −0.7 (*Figure 2D*). Thus, inferences conducted on subsamples of the same phylogenetic tree are partly independent. The precise degree of independence is expected to depend on the shape of the phylogenetic tree, but the coefficient was similar when doing the same study on a randomly generated tree instead of the SARS-CoV-2 tree.

We finally conducted another round of simulations to evaluate the error on what we considered as exchange between multiple locations when using sparse subsampling. For that, a 1,000,000-leaves tree was simulated with a five-states discrete trait representing geographical units. Then, 100 subsampled 1000-leaves trees from the whole phylogenetic tree were produced and the ancestry for the discrete trait was reconstructed from the leaf data only. We estimated the number of transitions (exchanges) of each type and compared them with the one obtained from the main tree, finding a mean error rate of 2.7% over the 100 subsamples (*Figure 2—figure supplement 1*).

Altogether, these simulations suggested that using subsamples of 1000 sequences from a large dataset and performing partially independent replicates seems to be sufficient to accurately estimate transition events.

### Description of the datasets and global diversity of SARS-CoV-2 sequences

We defined 100 subsamples of sequences proportionally to COVID-19 deaths across geographic locations and time for different geographic scales (worldwide, Europe, and French regions) and time

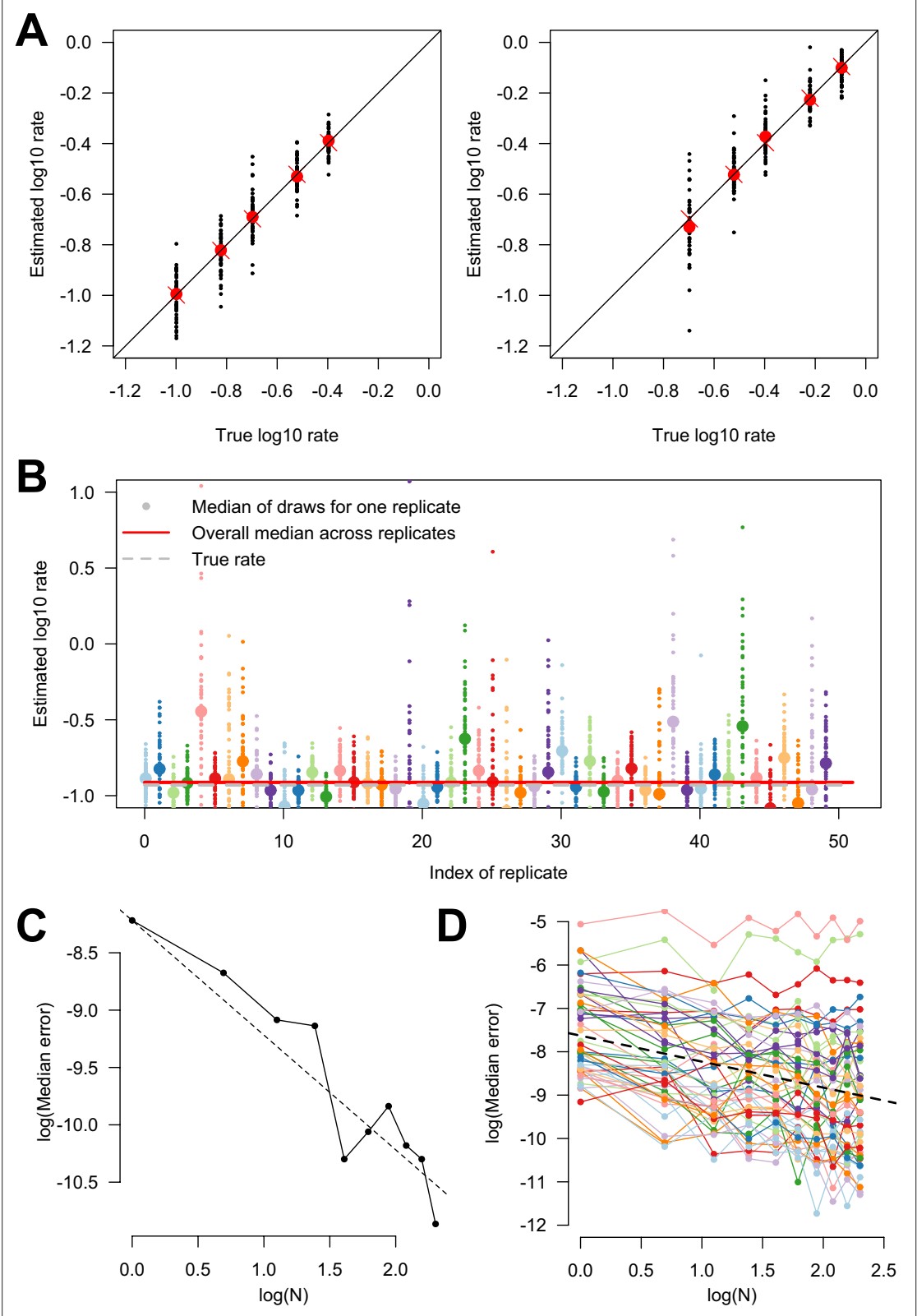

**Figure 2.** Estimating variability in transition rates using simulations. (**A**) Estimated versus true parameters in the simulation study of the two-states model. The two panels show the two transition rates. For each set of parameters, 50 replicates were conducted. The large red dot is the median of the replicates. The red cross is the true parameter value, on the bisector. (**B**) Estimated rate of transition in subsampled trees. For each replicate (*n* = 50), one point is the result of one subsampled smaller phylogenetic tree (from a large phylogenetic tree). The big dot shows the median for each replicate.

*Figure 2 continued on next page*

*Figure 2 continued*

The horizontal red line is the overall median (of the medians), across replicates. The horizontal dashed gray line is the true rate. Only one of the two rate parameters is shown. (**C**) Log-median error in parameter estimation as a function of the log number of replicates, when inference is conducted on truly independent replicate evolutionary histories, on a tree of 1000 leaves. The points are the data, the dashed line shows the line of slope '−1' which is the expectation as the replicates are truly independent. (**D**) Log-median error as a function of log number of subsamples used for the inference done on subsampled phylogenetic trees. The colored points and lines show the inference done on 50 distinct realizations of the evolutionary process on the whole tree. The dashed line is the overall regression line with a slope of −0.7. We tested between 1 and 10 subsampled trees (*x*-axis).

The online version of this article includes the following figure supplement(s) for figure 2:

**Figure supplement 1.** Mean error proportion of discrete trait transitions when subsampling a bigger tree.

**Figure supplement 2.** Positive correlation between the number of severe acute respiratory syndrome coronavirus 2 (SARS-CoV-2)-related deaths and the number of sequences included within a subsample on (**A**) worldwide, (**B**) European, and (**C**) French scales.

**Figure supplement 3.** Differential profiles of severe acute respiratory syndrome coronavirus 2 (SARS-CoV-2) sequences wanted and available for each French administrative region.

**Figure supplement 4.** Within-territory pairwise patristic distance across the subsamples for each period on (**A**) worldwide, (**B**) European, and (**C**) French administrative region scales.

periods (from January 1 to July 25, 2020, and from July 26 to December 31, 2020, respectively, covering the first and second European epidemic waves). We chose the sampling intensity guided by the weekly number of SARS-CoV-2-related deaths reported by public health organizations. Here, the number of SARS-CoV-2-related deaths was used rather than the number of detected cases because the latter was biased due to variable ascertainment rates across countries and time. For example, the larger number of PCR tests conducted in the second epidemic wave could wrongly suggest that the virus circulated much more during the second half of 2020 (*Figure 1—figure supplement 2*).

For each geographic scale and time period, there was a positive correlation between the weekly number of SARS-CoV-2-related deaths and the weekly number of sequences we included for a subsample (Spearman's rank correlation, p < 0.001; *r* = 0.94 for the lowest correlation). We also confirmed that the number of sequences per territory was, on average, properly temporally distributed within each time period (*Figure 2—figure supplement 2*).

Some countries and French administrative regions were however discarded in the analyses because they were not sufficiently represented in the GISAID database. Overall, a total of 39,288 and 39,755 distinct SARS-CoV-2 sequences were included across the 100 sampled phylogenies for the worldwide dataset, respectively, for the first and the second time periods (*Table 1*). At the European scale, 26,757 and 27,658 different SARS-CoV-2 sequences covering 11 countries were analyzed across the 100 subsamples (*Table 1*). Focusing on French administrative regions, sequences available on the GISAID database were very sparse. The Provence-Alpes-Côte d'Azur (PACA, Marseille area) was the

**Table 1.** Number of severe acute respiratory syndrome coronavirus 2 (SARS-CoV-2) sequences investigated for each dataset.

| Dataset | Geographies | Period investigated | Average number of sequences sampled in a subsample | Total number of sequences |
|---|---|---|---|---|
| World | Africa, Asia, Europe, France, North America, Oceania, South America | January 1 to July 25, 2020 | 846 | 39,288 |
| | | July 26 to December 31, 2020 | 777 | 39,755 |
| Europe | Belgium, France, Germany, Italy, The Netherlands, Poland, Romania, Russia, Spain, Sweden, United Kingdom | January 1 to July 25, 2020 | 904 | 26,757 |
| | | July 26 to December 31, 2020 | 872 | 27,658 |
| France | Auvergne-Rhône-Alpes (ARA), Bretagne (BRE), Île-de-France (IDF), Occitanie (OCC), Provence-Alpes-Côte d'Azur (PACA) | January 1 to July 25, 2020 | 416 | 2543 |
| | | July 26 to December 31, 2020 | 433 | 3124 |

only region that highly sequenced SARS-CoV-2 in 2020. Île-de-France (IDF, Paris area), Auvergne-Rhône-Alpes (ARA, Lyon area), Occitanie (OCC, Toulouse and Montpellier area), and Bretagne (BRE, Rennes area) have sequenced much less than PACA, but provided sufficient data to investigate SARS-CoV-2 geographic exchange events in France. The remaining French administrative regions were discarded since too few sequences were available to properly match the number of weekly SARS-CoV-2 deaths (*Figure 2—figure supplement 3*). We thus considered 2543 unique sequences across the 100 subsamples between January 1 and July 25, 2020, and 3124 unique sequences between July 26 and December 31, 2020 (*Table 1*).

The genomic diversity of circulating SARS-CoV-2 in the different continents, countries, and French regions was found to be similar (*Figure 2—figure supplement 4*). Overall, genomes showed high sequence conservation compared to the Wuhan-Hu-1 reference in 2020 (mean and median of ~13 single nucleotide polymorphisms (SNPs) with 95% of the distribution comprised between 4 and 25 SNPs).

## Which continents exchanged SARS-CoV-2 with Europe and France?

Through 100 distinct, dated and ancestrally reconstructed phylogenetic trees, we first studied SARS-CoV-2 exchanges worldwide for each of the time periods studied. Between January 1 and July 25, 2020 (covering the first European epidemic wave), we found that Europe (excluding France) accounted for 57.3% of the total number of exportation events, and was the main source of SARS-CoV-2 exportations toward the other continents in all of the subsamples (*Figure 3A–D* and *Figure 3—figure supplement 1*). North America also highly participated in virus exportation during this period time (24.3%). South America and Asia were each associated with 7.1% of the total number of exportation events, consistent with a later circulation of the virus in these continents (*Figure 1—figure supplement 1*). France was estimated to have contributed 4.2% of the total exportation events, indicating that France was not the major European source of SARS-CoV-2 at the international level between January 1 and July 25, 2020. The exportation events from France were mostly headed toward Europe and, to a lesser extent, to North America, South America, and Asia (*Figure 3B* and *Figure 3—figure supplement 2*). These events mostly consisted of the B.1 (80.2%), B.1.1 (6.1%), and B.1.356 (3.5%) lineages (*Figure 3E*). North America received a large proportion of SARS-CoV-2 from other continents (28.5% of the introduction events), followed by South America (23.1%) and Europe (17.3%) (*Figure 3A–D*). An average of 11.7% of all SARS-CoV-2 introductions were into France, and originated from North America (50.7%) and Europe (45.7%) (*Figure 3—figure supplement 2*). These introductions consisted of the B.1 (71.9%) and B.1.1 (18.2%) lineages (*Figure 3E*). The first introductions into France were detected at the beginning of February, and progressively increased to reach a peak just before the nationwide lockdown from March 17, 2020 (*Figure 3D*). Only South America and Asia were associated with a continuous increase in SARS-CoV-2 introductions after this date, probably because no such drastic measures were generalized there and the circulation of the virus remained limited in these regions.

From July 26 to December 31, 2020 (second European epidemic wave), we observed 1.4 times fewer exchange events worldwide compared to the first half of 2020. Importantly, we showed the importance of analyzing several subsamples, as there was a large variation in the total number of exportation or introduction events, especially in Europe (*Figure 3A*). Europe was, as between January 1 and July 25, 2020, the main source of exchanges with a total of 49.9% of the exportation events across subsamples, followed by North America (18.0%), Asia (14.1%), and South America (9.5%) (*Figure 3A–D* and *Figure 3—figure supplement 1*). Most of the events occurred during the summer period (June to August 2020), corresponding to the summer holidays in most countries of the world. France accounted for 8.3% of the exportation events, but they were almost exclusively oriented toward other European countries (89.9%) and overall detected from August to November 2020 (*Figure 3B* and *Figure 3—figure supplement 2*), consistent with the SARS-CoV-2 incidence in this period in France (*Figure 1—figure supplement 2*). The B.1.160 lineage accounted for almost all the exportation events from France (97.9%) (*Figure 3E*). In a similar fashion, SARS-CoV-2 introductions into France mostly originated from Europe (81.9%) (*Figure 3B*) and were detected at a low rate from April 2020, then at a higher but always limited rate from June 2020, and at a strong level in September and October 2020 (*Figure 3—figure supplement 2*). These SARS-CoV-2 introductions into France consisted in majority of B.1.177 (28.0%), B.1.160 (24.7%), B.1 (17.7%), B.1.1 (10.0%), and B.1.258 (7.6%) lineages (*Figure 3E*).

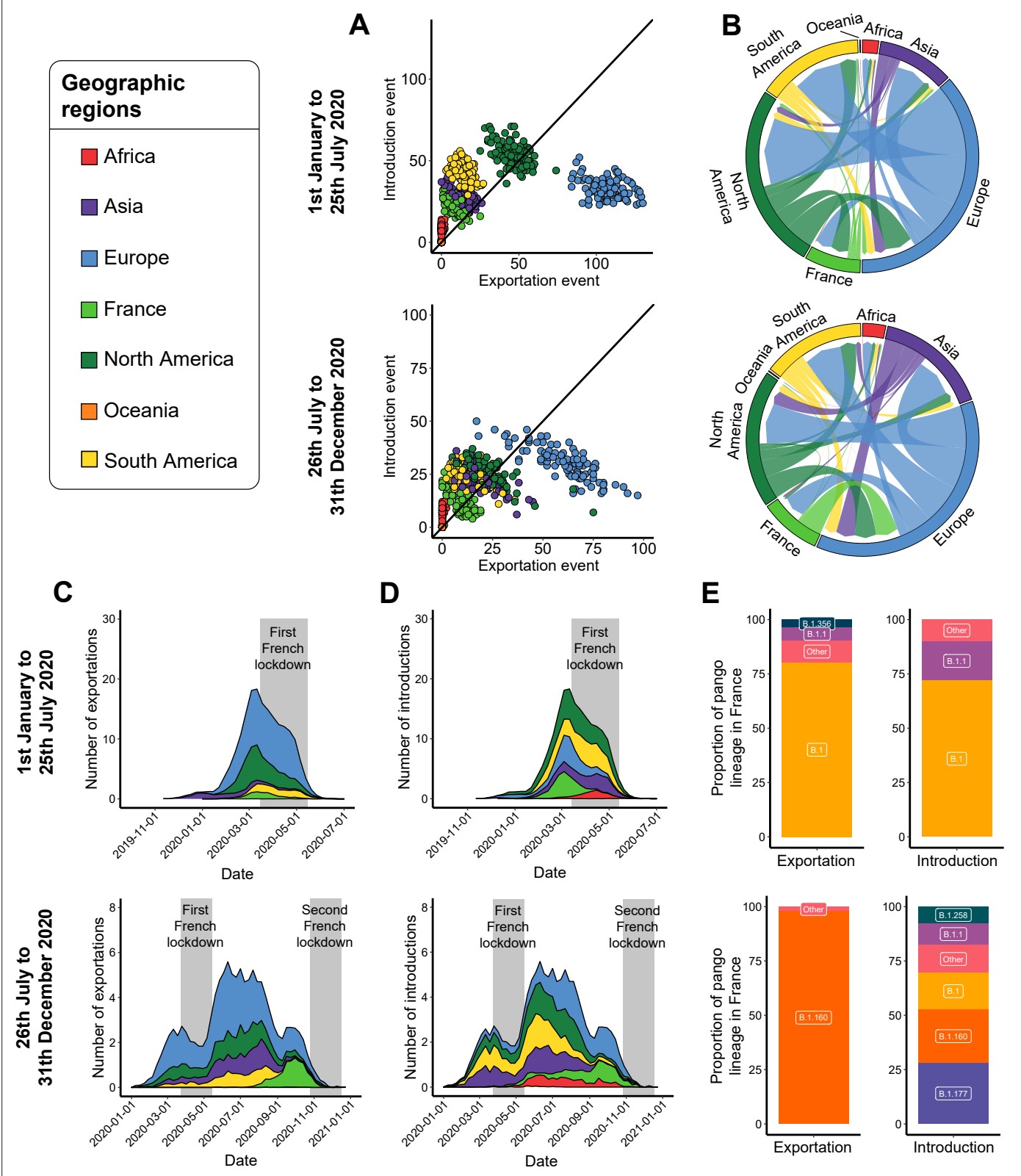

**Figure 3.** Severe acute respiratory syndrome coronavirus 2 (SARS-CoV-2) exchange worldwide. Exchange events were inferred with 100 subsampled phylogenies between January 1 and July 25, 2020, and between July 26 and December 31, 2020. (**A**) Number of introduction and exportation events for each subsample and for each continent and France. (**B**) SARS-CoV-2 exchange flows between continents and France during the two time periods investigated. In these plots, migration flow out of a particular location starts close to the outer ring and ends with an arrowhead at the destination

*Figure 3 continued on next page*

Figure 3 continued

location. Arrow width is proportional to the exchange strength. (**C**) Number of exportation and (**D**) introduction events per territory over time. The mean number of exchanges over the subsamples and for each week was calculated. Gray bars indicate the dates of the complete lockdowns in France. (**E**) Proportion of pango lineages exported from France and introduced into France. Lineages with a proportion <3% were grouped into the 'other' clade.

The online version of this article includes the following figure supplement(s) for figure 3:

**Figure supplement 1.** Severe acute respiratory syndrome coronavirus 2 (SARS-CoV-2) exchanges at the worldwide scale across subsamples.

**Figure supplement 2.** Introduction and exportation events from the perspective of France at the worldwide scale.

## How did the virus spread in Europe?

We then aimed to get a more comprehensive view of SARS-CoV-2 exchanges between France and other European countries with the same approach. Here, we only focused on European countries associated with a high incidence and without under-sampling due to a lack of data on GISAID (*Table 1*).

By calculating the count of introduction and exportation events between January 1 and July 25, 2020 across the subsamples, we observed that Italy was the major contributor to virus exportation toward other European countries, with an average of 41.5% of the total number of exportation events. The United Kingdom, France, and Spain also highly participated in virus exportation, and consisted of 21.6, 18.1, and 11.8% of the total number of exportation events, respectively (*Figure 4A–D* and *Figure 4—figure supplement 1*). These observations are in line with epidemiological data, since Italy was the first country in Europe to be heavily affected by the pandemic; and France, the United Kingdom, and Spain were the three other European countries associated with the highest number of SARS-CoV-2-related deaths during the first wave (*Figure 1—figure supplement 1*). The number of all exportation events however decreased after the implementation of lockdowns in the different countries (with the first one occurring in Italy on March 9, 2020) (*Figure 4C*). France mostly exported SARS-CoV-2 toward Belgium (25.5%), Germany (21.0%), and the United Kingdom (20.4%), and a little less toward Spain (10.1%) and the Netherlands (9.2%). All of these events occurred before the official lockdown in France (March 17, 2020) (*Figure 4B* and *Figure 4—figure supplement 2*), and consisted of the B.1 (78.5%), B.1.1 (8.8%), and B.1.356 (6.9%) lineages (*Figure 4E*). The rate of SARS-CoV-2 exportations from France then decreased until the second European epidemic wave, as it was also the case for other European countries except Russia (*Figure 4C*). For all introduction events, the proportions were more balanced: the United Kingdom accounted for a quarter of the total number of events, while Russia, Belgium, Germany, Italy, Spain, France, and the Netherlands represented between 6.5 and 11.9% of the total number of events (*Figure 4D*). In France, a high rate of introduction events was observed in February and March before the lockdown and originated mostly from Italy (44.3%), the United Kingdom (30.8%), and Spain (14.6%) (*Figure 4—figure supplement 2*). These introductions consisted in majority of the B.1.1 (47.9%) and B.1 (35.4%) lineages (*Figure 4E*).

The second time period (from July 26 to December 31, 2020) showed a different pattern of exchanges. Here, we estimated 1.3 times fewer exchanges compared to the first half of 2020. Russia accounted for most of the exportation events (27.6%) (*Figure 4A, B*). These events were estimated to occur during the spring (after the relaxation of containment measures in most European countries) and the summer periods (*Figure 4C*). This result was expected since Russia was almost the sole European country to report a high number of SARS-CoV-2-related deaths during this period (*Figure 1—figure supplement 1*). Spain (16.9%), France (14.0%), Germany (10.2%), Italy (7.9%), Poland (7.5%), and the United Kingdom (6.6%) also highly participated in virus exportation (*Figure 4A–D* and *Figure 4—figure supplement 1*). Most of these events were detected between August and October 2020 (*Figure 4C*), and strongly decreased just before the second lockdown in most European countries (the first one occurring in Spain on October 9, 2020). Again, these observations are consistent with epidemiological reports, as Spain was the first country in European Union to be associated with a sharp increase of SARS-CoV-2-related deaths, rapidly followed by France. France mostly exported the virus toward Italy (22.2%), Germany (21.3%), and Belgium (19.1%) (*Figure 4B* and *Figure 4—figure supplement 2*), and mostly the B.1.160 lineage (79.2%) (*Figure 4E*). Focusing on introduction events, Germany accounted for 23.8% of the total number of introductions, followed by the United Kingdom (15.3%), Italy (15.1%), and France (9.8%). For the remaining European countries, the proportion of introduction events was comprised between 4.5 and 7.6%, except for Russia that accounted for <0.1% of the events. Introductions into France mostly originated from Russia (31.7%) during the summer

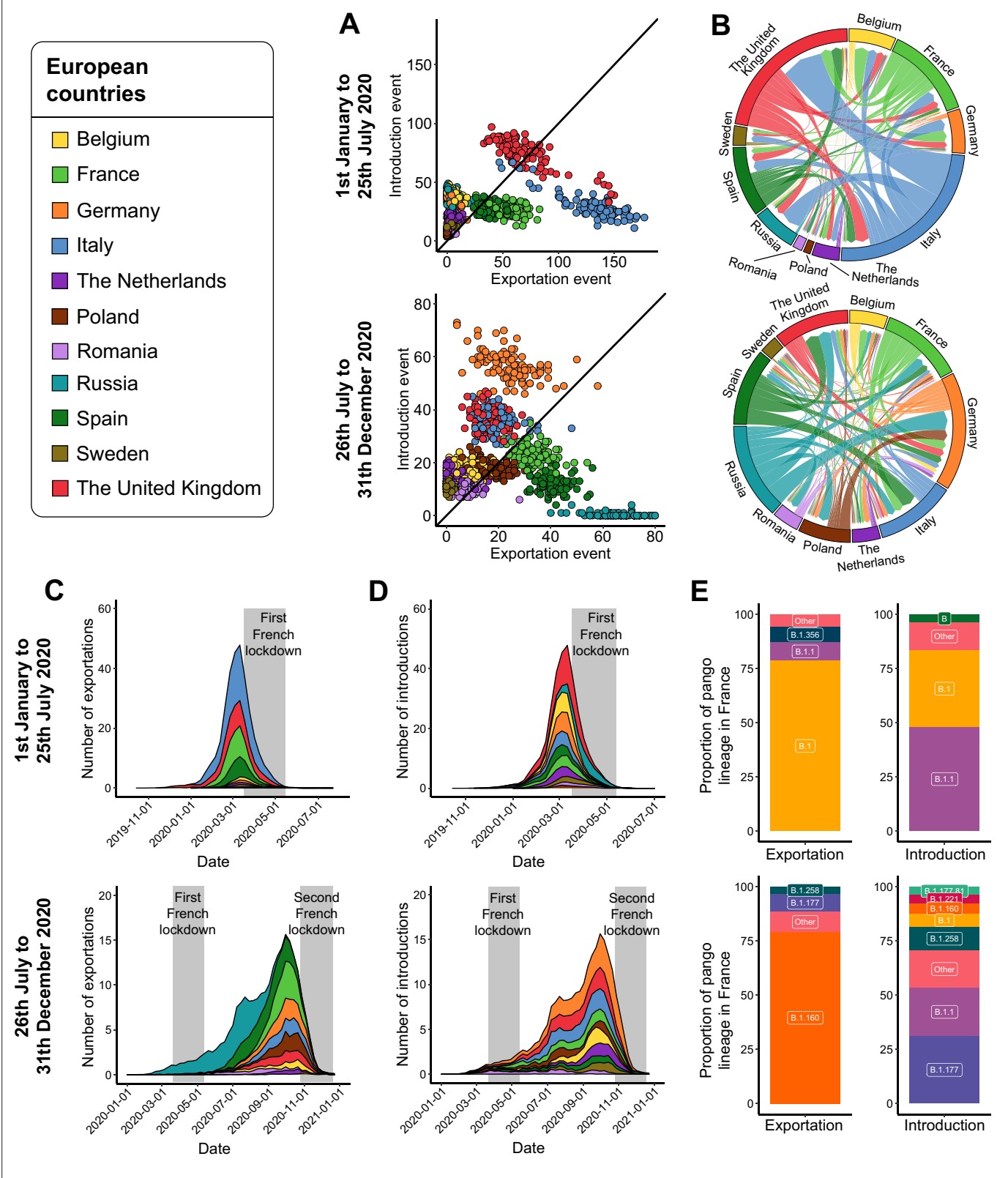

**Figure 4.** Severe acute respiratory syndrome coronavirus 2 (SARS-CoV-2) exchanges on the European scale. Transmission events were calculated by averaging the results from 100 subsampled phylogenies between January 1 and July 25, 2020, and between July 26 and December 31, 2020. (**A**) Number of introduction and exportation events for each subsample and for each European country. (**B**) SARS-CoV-2 exchange flows between European countries during the two time periods investigated. In these plots, migration flow out of a particular location starts close to the outer ring and ends with

*Figure 4 continued on next page*

*Figure 4 continued*

an arrowhead at destination location. Arrow width is proportional to the exchange strength. (**C**) Number of exportation and (**D**) introduction events per territory over time. The mean number of exchanges over subsamples and for each week was calculated. Gray bars indicate the dates of the complete lockdowns in France. (**E**) Proportion of pango lineages exported from France and introduced into France. Lineages with a proportion <3% were grouped into the 'other' clade.

The online version of this article includes the following figure supplement(s) for figure 4:

**Figure supplement 1.** Severe acute respiratory syndrome coronavirus 2 (SARS-CoV-2) exchanges on the European scale across subsamples.

**Figure supplement 2.** Introduction and exportation events from the perspective of France on the European scale.

period, and later from Spain (23.7%), Italy (12.5%), and Germany (11.0%) (*Figure 4—figure supplement 2*). These SARS-CoV-2 introductions into France consisted in majority of B.1.177 (31.0%), B.1.1 (22.6%) and B.1.258 (7.6%) lineages (*Figure 4E*).

## The first and second French epidemic waves: two patterns of SARS-CoV-2 transmissions within France

We finally conducted an analysis of virus spread inside of France by studying exchanges between French administrative regions. As previously explained, only five regions (ARA, BRE, IDF, OCC, and PACA, which cover the most densely populated urban areas in France, that is, Paris, Lyon, Montpellier, Toulouse, and Marseille) were investigated since only a few, if any, sequences were available on GISAID for the other regions (*Figure 2—figure supplement 3*). Of note, the Grand-Est region was one of the first massively affected by SARS-CoV-2 in France during the first wave and, thus, must have played a major role. The fact that it was not sampled is a limitation that must be taken into account when reading these results.

During the first French epidemic wave, almost all exportation events originated from IDF (84.0%). ARA, BRE, OCC, and PACA only accounted for 8.2, 0.4, 2.3, and 5.1% of the total number of exportation events (*Figure 5A, B* and *Figure 5—figure supplement 1*). As for the wider geographic scales, most exportation events occurred between February and March, and largely decreased after the national lockdown (*Figure 5C* and *Figure 5—figure supplement 2*). ARA was the region having most SARS-CoV-2 introductions, with 48.3% of the total number of introduction events, followed by PACA (22.1%), OCC (17.3%), and BRE (7.0%) (*Figure 5C*). All the regions were mostly affected by the B.1 lineage at similar rates, except for OCC with more B.1.1 lineage (*Figure 5E*).

During the second wave, IDF was no longer the single epicenter of SARS-CoV-2 inter-region exchanges. Both IDF and ARA participated concomitantly in virus spread at similarly large levels (43.3 and 42.6% of the total number of exportation events, respectively) (*Figure 5A, B* and *Figure 5—figure supplement 1*). The first exportation events mostly occurred in the spring and summer in IDF. ARA was associated with a high increase in exportation events by the end of July, consistent with the resumption of the epidemic. Again, the second national lockdown – implemented in October 29, 2020 – was associated with a strong decline of the number of exchange events (*Figure 5D*). Regarding introduction events, PACA received 31.3% of the total number of introductions, followed by ARA (24.7%), IDF (24.1%), OCC (11.5%), and BRE (8.2%). Overall, ARA, BRE, OCC, and PACA mostly sent and received the B.1.160 lineage. Only IDF showed a different pattern, exporting at similar rates both the B.1.160 and the B.1.177 lineages (*Figure 5F*).

Altogether, SARS-CoV-2 exchange events within France showed two distinct patterns between the first and second epidemic waves of 2020. While most SARS-CoV-2 exchanges involved the Paris area during the first half of 2020, the situation was much more balanced after the summer 2020 with the Lyon area (ARA) also contributing to many exchanges. Viral spread across French regions in summer 2020 could be explained by travel associated with the summer holidays.

## Discussion

Prior to this analysis, the dynamics of SARS-CoV-2 transmissions within France or between France and other countries (in Europe or worldwide) has been only partially characterized. To our knowledge, three studies have explored the virus spread in France, but only focused on the first French epidemic wave and/or could not address SARS-CoV-2 introduction events from other continents or other European

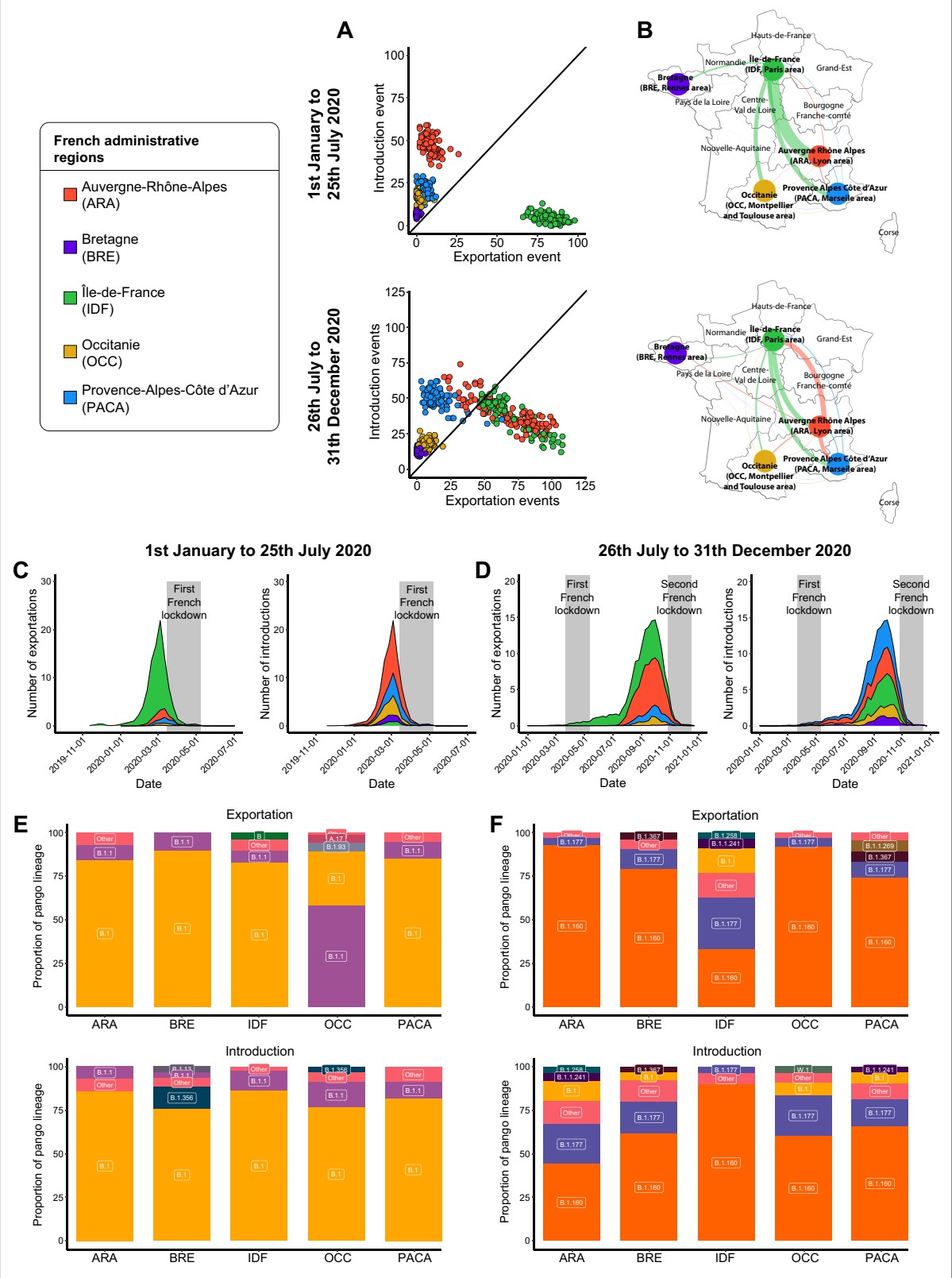

**Figure 5.** Severe acute respiratory syndrome coronavirus 2 (SARS-CoV-2) exchanges in France. Exchange events were calculated through 100 subsampled phylogenies between January 1 and July 25, 2020, and between July 26 and December 31, 2020. (**A**) Number of introduction and exportation events for each subsample and for each French administrative region. (**B**) SARS-CoV-2 exchange flows between French administrative regions during the two time periods investigated. Migration flow out correspond to the mean of the French sets of sequences. The number of

*Figure 5 continued on next page*

Figure 5 continued

exportation and introduction events per French region over time are detailed (**C**) between January 1 and July 25, 2020, and (**D**) between July 26 and December 31, 2020. The proportion of pango lineages exported or introduced are given for each French region during (**E**) the first and (**F**) the second time period investigated. Lineages with a proportion <3% were grouped into the 'other' clade.

The online version of this article includes the following figure supplement(s) for figure 5:

**Figure supplement 1.** Severe acute respiratory syndrome coronavirus 2 (SARS-CoV-2) exchanges on the French scale across subsamples.

**Figure supplement 2.** Introduction and exportation events from the perspective of each French administrative region.

countries (*Danesh et al., 2020*; *Elie and Alizon, 2020*; *Gámbaro et al., 2020*). Other studies have focused on larger scales such as Europe as a whole, but did not explore in-depth the dynamics of exchanges especially in France (*Hodcroft et al., 2021*; *Lemey et al., 2021*; *Nadeau et al., 2021*). Here, we studied introduction and exportation events in France at different scales during the first year of the pandemic, covering the two first European epidemic waves. Our approach provides a way to analyze large amounts of genomic data for future epidemics and improve our understanding of viral spread in France and Europe.

Methodologically, most phylodynamics studies on SARS-CoV-2, as well as on other viruses or pathogens, are based on Bayesian approaches (notably with the popular and robust BEAST tool; *Drummond and Rambaut, 2007*) using one or more phylogenetic trees composed of several thousand sequences. To reconstruct the evolution parameters and ancestral states at nodes (here, the geographical area and the SARS-CoV-2 lineage), Bayesian methods require high computational power, representing their main limitation (*Drummond and Rambaut, 2007*). Overall, the generation of a dated phylogenetic tree and the ancestral state reconstruction for even a few thousands of sequences is no small feat, requiring a very long computational time, and sometimes parameter estimation fails to converge. Furthermore, conclusions obtained would be based on a single sample with a relatively limited number of sequences, but different samples may lead to different results (*Hall et al., 2016*). As stated by Dellicour et al., Bayesian phylogenetic analysis on about 2800 SARS-CoV-2 sequences required over 150 hr to obtain enough samples from the posterior distribution across 15 parallel runs with a GPU accelerated implementation (*Dellicour et al., 2021*). Since the GISAID database stores several hundred thousand SARS-CoV-2 sequences ($n$ = 638,704 on May 8, 2022 for the year 2020 alone), we believe that using a phylogeny of a few thousand sequences only might not be the best way to accurately infer exchange events. In order to obtain more comprehensive trends drawing conclusions from a larger number of SARS-CoV-2 sequences, we constructed 100 semi-independent maximum likelihood phylogenies of relatively limited size (from 418 to 904 sequences depending on the geographic scale and time period investigated) and inferred ancestral states by a maximum likelihood discrete trait phylogeographic method. Bayesian approach seems not to be more accurate than maximum likelihood for the reconstruction of ancestral states (*Hanson-Smith et al., 2010*), and we gain a lot in terms of sampling, computational time, and trends by sampling 100 semi-independent subsamples. The maximum likelihood-based method was recently used to depict with accuracy the genomic epidemiology of the virus in Canada (*McLaughlin et al., 2022*), as well as in other contexts (*Cunningham et al., 1998*; *Pupko et al., 2000*).

We characterized SARS-CoV-2 spatial spread within France, and between France and European and all other countries in 2020. Between January 1 and July 25, 2020 (first European epidemic wave), France mostly exchanged SARS-CoV-2 with Europe and North America. The observation contrasts with the second European epidemic wave – that took place between late October and the end of 2020 – where France almost exclusively exchanged the virus with other European countries. This result is logical insofar as the borders of France have gradually reopened with European countries but remained mostly closed to the rest of the world. A very low rate of SARS-CoV-2 exchanges between Europe (including France) and Asia or South America was found in 2020, which is not surprising since the virus started to spread in these continents only from May 2020 where all borders of Europe were already closed. On the European scale, France was a major source of SARS-CoV-2 exchange. It ranked third regarding exportation events (mainly directed toward neighboring countries) only behind the United Kingdom and Italy between January 1 and July 25, 2020, and third again behind Spain and Russia (where exchange events happened mostly in summer 2020) between July 26 and December 31, 2020. The first cases introduced from European countries into France came from Italy and Spain,

consistent with the early days of SARS-CoV-2 spread in Europe. These results are in line with the observations made across Europe through Bayesian inference (*Lemey et al., 2021*). In line with the lack of SARS-CoV-2 genomic diversity at the beginning of the pandemic, both B.1 and B.1.1 were the most representative lineages introduced into or exported from France. During the second European epidemic wave, some cases were introduced into France (and some other European countries) from Russia where virus incidence was high since the summer 2020. Then, most introductions into France originated from Spain which initiated the second wave in European Union, followed by Italy and Germany. Most of the introductions into France were the lineages B.1.1, B.1.177, and B.1.258 that highly circulated in Europe. By contrast, France mostly exported the B.1.160 lineage which was firstly detected in July 2020 in Marseille (PACA, France), and represented the majority of sequences in fall 2020. The frequency of the B.1.160 lineage then decreased rapidly at the beginning of 2021 in favor of B.1.1.7. Interestingly, the number of international exchanges was lower during the second wave compared to the first one, indicating the effectiveness of border control measures and that French people reduced travel outside France after the first national lockdown.

Our results on the French scale should be interpreted considering several limitations. In addition to a limited overall size, genomic data only covered 5 out of 13 administrative regions (*Figure 2—figure supplement 3*). The North and East of France (i.e., regions Hauts-de-France and Grand-Est, respectively) that border Belgium, Luxembourg, and Germany, were not investigated. More importantly, the region Grand-Est was associated with early large clusters that participated in virus spread across the country during the first European epidemic wave, such as the evangelical gathering of the Christian Open-Door Church (Mulhouse, Haut-Rhin) that took place from 17 to February 21, 2020 and consisted of approximately a thousand infected individuals. Keeping these limitations in mind, we observed two distinct patterns of SARS-CoV-2 exchanges within France in 2020. During the first wave, most of the exportations originated from IDF (Paris area). This may be linked with the larger number of individuals living in IDF who left to other French regions at the beginning of the epidemic. In the second epidemic wave, we observed a similar contribution of IDF to exchanges, but this time accompanied with ARA (Lyon area) which was associated with a strong SARS-CoV-2 incidence. The circulation of lineages was the same between the French regions, both during the first and second epidemic waves. The B.1 was the major circulating lineage during the first wave, while B.1.160 was mostly transmitted during the second one, concomitantly with B.1.177 in IDF.

To conclude, using an original phylodynamic approach to investigate a large SARS-CoV-2 sequence dataset, we characterized SARS-CoV-2 spatial spread within France, across Europe and in the world in 2020 and unraveled distinct patterns of exchanges between the two first epidemic waves.

## Materials and methods

### Sequence acquisition and curation, and dataset production

A phylodynamics study (which aims to depict with accuracy exchange events between locations) requires a large number of sequences (*Attwood et al., 2022*). SARS-CoV-2 genome sequences were downloaded from the GISAID database on May 8, 2022 ($n = 10,005,024$). The following inclusion and exclusion criteria were applied to select samples for analysis: (1) only complete viral genomes from infected individuals (more than 29 kb in length) were included; (2) the maximal proportion of undetermined nucleotide bases was fixed to 5%; (3) sequences with unknown or incomplete sampling dates, or unknown geographical location, were excluded; (4) sampling dates were comprised between January 1, 2020 and December 31, 2020; and (5) sequences from non-human were excluded. The final dataset included 638,704 sequences. In a second pass, 26,103 genomes were considered as outliers as they were in the extreme 5% regarding the number of non-synonymous mutations.

We studied the epidemic at three different geographic scales: (1) worldwide, to identify SARS-CoV-2 exchanges between continents and France; (2) at the European scale, to determine more precisely which European countries mostly exchanged with France; and finally (3) at the French administrative region level, to get a better understanding of virus spread inside the country. For each geographic scale, the analyses were independently run for the two first European epidemic waves (January 1 to July 25, 2020 (weeks 1–30), and July 26 to December 31, 2020 (weeks 31–52); *Figure 1*).

Considering the large number of sequences in GISAID that passed our criteria ($n = 638,704$), we produced 100 subsamples of sequences for each investigated geographic scale analysis (i.e.,

worldwide, Europe, and French administrative regions) and time period (January 1 to July 25, 2020 and July 26 to December 31, 2020). To determine the number of SARS-CoV-2 genome sequences to be included so as to be representative of each location epidemic, we used the number of SARS-CoV-2-related deaths per territory and per week as a proxy for virus circulation taking place 2 weeks earlier. We did not use SARS-CoV-2 detected cases as the proxy for infections because the intensity of COVID-19 testing was increased after the first wave, which probably increased the ascertainment probability (*Figure 1—figure supplement 2*). For each subsample, sequences were drawn randomly from our GISAID extraction. Some countries and French regions had to be discarded because of their under-representation in the GISAID database. For the European scale, we selected countries with more than 10,000 sequences. Ukraine, Czech Republic, and Hungary passed this threshold, but were excluded for having too many weeks without enough data. At the French region scale, we included only regions with less than 20% of weeks lacking data (*Figure 2—figure supplement 3*).

For the first time period investigated (January 1 to July 25, 2020), an average of 846, 904 and 416 SARS-CoV-2 sequences were included in each set at the worldwide, European and French scales, respectively (*Table 1*). When considering the second time period (July 26 to December 31, 2020), the worldwide, European and French scale sequence subsamples contained 777, 872, and 433 genomic sequences (*Table 1*).

## Defining appropriate subsample using simulations

Before performing phylodynamics analysis using SARS-CoV-2 phylogenetic trees of limited sizes, we performed a series of simulations to demonstrate the accuracy of our approach. In a first simulation, we investigated the inference of transition parameters on a random phylogenetic tree of 1,000 leaves produced with the *rtree* function of the ape package (*Paradis et al., 2004*) in R (*R Development Core Team, 2022*). For that, we simulated a two-states model, with two different transition rates, using the ape *rTraitDisc* function (*Paradis et al., 2004*) with the 'all rates different' (ARD) model. From each simulation, the two transition parameters were inferred using the *ace* function of the ape package (which performed maximum likelihood inference of the two rates). In this simulation, we tested different combinations of the two rates.

In a second simulation analysis, we tested the accuracy of inference done on subsamples of a larger phylogenetic tree. We considered a SARS-CoV-2 phylogenetic tree of 10,000 leaves retrieved from the GISAID database, and subsampled 50 randomly chosen 100-leaves phylogenies for inference. For each resulting subtree, we conducted inferences – using the same strategy as previously explained – on 50 replicates corresponding to 50 realizations of the stochastic process of evolution of the discrete character on the whole tree of 10,000 leaves.

Then, we evaluated the level of independence between estimations done on subsampled trees, using the SARS-CoV-2 phylogenetic tree of 10,000 leaves and following the same strategy as previously detailed.

In order to evaluate the error on counting transitions between states of a discrete trait along a phylogeny when using sparse subsampling, we simulated a main tree of 1,000,000 leaves with a five-states discrete trait again using *rtree* and *rTraitDisc* (ARD model) from the R ape package (*Paradis et al., 2004*). We then made 100 random subsampled 1000-leaves trees and reconstructed the ancestry for the discrete trait from the leaf data only using the ape function *ace* (once again ARD model). The mean error rate was obtained by calculating the number of transitions of each type and comparing them with the one obtained from the main tree. The non-parametric 95% confidence interval was calculated from 1000 bootstraps.

## Multiple sequence alignment and maximum likelihood phylogenetic tree

For each subsample, genome sequences were independently aligned against the Wuhan-Hu-1 reference genome (GenBank accession: NC_045512.2) using MAFFT v7.450 then merged altogether (*Katoh and Standley, 2013*). The alignments were then masked for problematic sites listed by Nicola De Maio et al. (https://virological.org/t/masking-strategies-for-sars-cov-2-alignments/480). All the sites present in the list were transformed as 'N' and thus eliminated from the subsequent analyses. From each masked alignment, we inferred a maximum likelihood phylogenetic tree using FastTree v.2.1.11 under a general time-reversible (GTR) model with a discrete gamma distribution to model

inter-site rate variation and 1000 ultrafast-bootstraps to compute support values (*Price et al., 2010*). The choice of FastTree was mainly due to its short runtime while giving results very close to IQTree 2.1.2 (*Minh et al., 2020*) as described by Rob Lanfear (https://github.com/roblanf/sarscov2phylo/blob/master/tree_estimation.md). Each phylogenetic tree was dated using LSD2 v2.3 using Wuhan Hu-1 as outgroup for rooting (*To et al., 2016*), with the parameter '-e 2' to be more stringent about outlier nodes. Phylogenetic trees were manipulated and viewed using treeio and ggtree R packages, respectively (*Wang et al., 2020*; *Yu et al., 2017*).

## Worldwide, Europe, and French region SARS-CoV-2 phylodynamics analyses

For each phylogenetic tree we produced, ancestral state reconstructions of discrete geographical and SARS-CoV-2 lineage data were performed with the *ace* function of ape package in R (*Paradis et al., 2004*). Three distinct Markov models of discrete character evolution through maximum likelihood were tested and compared: the Equal Rates (ER) which assumes a single rate of transition among all possible states, the All Rates Different (ARD) which allows a distinct rate for each position transition between two states, and the Symmetrical Rates (SYM) where forward and reverse transitions share the same parameter. In this study, we only present the results obtained with the SYM model because both the ARD and ER models led to nearly identical observations compared to the SYM model. Of note, the use of ARD required extensive calculation times to infer maximum likelihood transition rates.

From the ancestral state reconstruction, we assigned nodes to a location when supported by ≥50% of the state likelihood. To detect and count each exchange event, we compared the geographic state of a node with the state of its two children. If the nodes corresponded to different geographic locations, we counted an exchange event from the parent node state to the given child node state. The midpoint of the parent–child branch was chosen as the date of exchange in all cases. The same strategy was applied to investigate the proportions of introduced or exported pango lineages. Exchange events across all the subsamples were then weekly averaged.

## Acknowledgements

We gratefully thank all the laboratories and organizations that deposited SARS-CoV-2 sequences on the GISAID database, without which this study would not have been possible. We would like to thank the editors of *eLife* and the reviewers, who provided extremely valuable feedback and greatly improved the quality of this work. Funding This study was supported in part by the ANRS MIE (Agence Nationale de la Recherche sur le SIDA et les hépatites virales – Maladies Infectieuses Emergentes), the FRM (Fondation pour la Recherche Médicale), and the Inserm UMR1137 unit.

## Additional information

### Funding
No external funding was received for this work.

### Author contributions
Romain Coppée, Conceptualization, Resources, Data curation, Software, Formal analysis, Validation, Investigation, Visualization, Methodology, Writing - original draft, Writing - review and editing; François Blanquart, Data curation, Software, Validation, Methodology, Writing - review and editing; Aude Jary, Valentin Leducq, Valentine Marie Ferré, Anna Maria Franco Yusti, Léna Daniel, Charlotte Charpentier, Samuel Lebourgeois, Karen Zafilaza, Vincent Calvez, Diane Descamps, Anne-Geneviève Marcelin, Writing - review and editing; Benoit Visseaux, Conceptualization, Supervision, Funding acquisition, Validation, Methodology, Project administration, Writing - review and editing; Antoine Bridier-Nahmias, Conceptualization, Resources, Data curation, Software, Formal analysis, Supervision, Validation, Investigation, Methodology, Project administration, Writing - review and editing

### Author ORCIDs
Romain Coppée (iD) http://orcid.org/0000-0002-3024-5928

François Blanquart [ID] http://orcid.org/0000-0003-0591-2466
Antoine Bridier-Nahmias [ID] http://orcid.org/0000-0002-0376-6840

**Decision letter and Author response**
Decision letter https://doi.org/10.7554/eLife.82538.sa1
Author response https://doi.org/10.7554/eLife.82538.sa2

## Additional files

### Supplementary files
• MDAR checklist

### Data availability
All genome sequences and associated metadata in the dataset are published in GISAID's EpiCoV database. To view the contributors of each individual sequence with details such as accession number, virus name, collection date, originating lab and submitting lab and the list of authors, visit: https://doi.org/10.55876/gis8.230120zd. All the scripts developed for this study were deposited in the following GitHub repository: https://github.com/Rcoppee/PhyloCoV (copy archived at swh:1:rev:9a984b6414e8c2377694ce922eb31874cdaddf1a).

The following dataset was generated:

| Author(s) | Year | Dataset title | Dataset URL | Database and Identifier |
|---|---|---|---|---|
| Coppée R | 2023 | EPI_SET_230120zd | https://doi.org/10.55876/gis8.230120zd | GISAID EpiCoVdatabase, 10.55876/gis8.230120zd |

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
