## [Editor Report]

This paper is a comprehensive, quantitative, and robust overview of the global, European, and French genomic epidemiology of SARS-CoV-2 in the first year of the pandemic. It contributes methodological advances in maximum likelihood phylogeography, using multiple scales and providing a simulation-based validation. The results show two distinct patterns of SARS-CoV-2 exchange events between the first and second half of 2020, with Europe being involved in most intercontinental exchanges: France experienced viral introductions primarily from North America and Europe during the first wave, while the second wave saw limited intercontinental movement and a significant contribution of the virus from Russia into Europe.

---

## [Decision Letter]

**Decision letter after peer review:**

Thank you for submitting your article "Phylodynamics of SARS-CoV-2 transmissions in France, Europe and the world during 2020" for consideration by *eLife*. Your article has been reviewed by 2 peer reviewers, and the evaluation has been overseen by a Reviewing Editor and a Senior Editor. The following individuals involved in review of your submission have agreed to reveal their identity: Christopher JR Illingworth (Reviewer #1); Angela McLaughlin (Reviewer #3).

As is customary in *eLife*, the reviewers have discussed their critiques with one another. What follows below is the Reviewing Editor's edited compilation of the essential and ancillary points provided by reviewers in their critiques and in their interaction post-review. Please submit a revised version that addresses these concerns directly. Although we expect that you will address these comments in your response letter, we also need to see the corresponding revision clearly marked in the text of the manuscript. Some of the reviewers' comments may seem to be simple queries or challenges that do not prompt revisions to the text. Please keep in mind, however, that readers may have the same perspective as the reviewers. Therefore, it is essential that you attempt to amend or expand the text to clarify the narrative accordingly.

Essential revisions:

Both reviewers and the Editor think that this is an important study that presents interesting findings about the spread of SARS-CoV-2 in France and in Europe, and are positive about the work's eventual publication. However, the reviewers brought up important considerations, including methodological concerns, that need to be addressed in a revised version. Following the consultation with the editor and reviewers, we would welcome a revised version that addresses the substantial methodological comments. In summary:

(1) Sampling: there are two questions. One is about the number of sequences and the number of deaths (both reviewers commented on this), as opposed to the number of infections; whether the correlation (sequences to deaths) is an artefact of the method used (Rev 1) and whether it validates the analysis (Rev2). In addition, since deaths and infections can be reported differently in different places, what is the impact of having different sampling fractions in different geographies? this could be addressed by looking at the robustness of the inferences to different choices of relative numbers of sequences from different regions

(2) Downsampling to < 1000 sequences, bootstrapping and adding: We were all concerned about the small individual datasets and the assumption that adding the results would generate the correct numbers of viral movements. After all, why this number of replicates – why not add after 5000 or 800 or some other number of replicates? Also, there are some instances where the same introduction event would be detected in many replicates, and somewhere it would be missed in most of them, so it's not clear that it's additive. Other authors have averaged, in similar analyses (but since each replicate might miss some events, averaging might not be). Given that this analytical method has the advantage that much larger numbers of sequences are tractable to analyze, the authors should analyze substantially larger datasets.

(3) The fast clock assumption has the potential to bias the analysis; a number of other related methodological issues were raised by reviewers. These should be addressed.

(4) Quantitative descriptions of the results and their variability should be included in the results (for example, the importation rate). This will aid in articulating the public health interpretation and relevance, which reviewers also note would add interest and weight to the paper.

*Reviewer #1 (Recommendations for the authors):*

The method is described as 'original' but line 377 implies that the same method was used to study data from Canada. Please clarify what is new about the approach used here relative to previous studies.

It is stated in line 65 that existing phylogenetic tools cannot include the large datasets generated by projects such as the international COVID sequencing effort. However, researchers on the COG-UK project generated a tree of sequences comprising hundreds of thousands, if not millions of sequences (https://www.cogconsortium.uk/priority-areas/data-linkage-analysis/public-data-analysis/). It would be helpful to clarify what can and cannot be achieved with which phylogenetic methods.

The positive correlations observed between the number of deaths and the number of sequences in a dataset (Figure 2) is a reassuring check but seems to be a direct artefact of the methods, whereby the number of deaths was used to determine the number of sequences. I suggest including these plots in supplemental material.

I would value more explicit information in the Methods section about how data from the inferences performed on different bootstrapped samples was combined.

The link to the stated GitHub repository did not work at the time I tried it.

*Reviewer #2 (Recommendations for the authors):*

L97: Figure comments: It is recommended to add an annotation for the date delineation of the two COVID waves analyzed in the main graphic. The legend should report how reproduction rate was calculated, and this should be clearly described in the methods. Relatedly, L447-448: state exact dates instead of 'mid' and 'late' months. Applies in results as well.

L106: Although the authors say that sequences are representative of 'sars-cov-2 infections', they then describe the comparison to deaths. Did the authors also compare case incidence to deaths and separately, to sequences over time? Need to better justify the use of deaths rather than new diagnoses. It would help the reader if there was also a presentation of cases over time in addition to deaths that could be done for example in Figure 1.

L114;L124;L209;L287,L465: Regarding the discarding of sequences from geographies that were underrepresented in the sequence set – how would the results be expected to change if these geographies were not discarded? They would be perhaps less likely to be sampled as well as represented in the ancestral nodes due to low representation, but they might have indicated additional nuances to the French regional transmission patterns.

Also, in the methods, what was the threshold by which the authors decided if a region was too underrepresented to be included? L404: what percent of cases and/or deaths were associated with the administrative regions that were excluded due to low or no sequences in each wave?

L115: Need to clarify how many sequences per phylogeny in the results. Were sequences sampled by geography randomly or proportionally to the relative number of deaths over time? Were the sampled sequences distributed uniformly over time or proportionally to deaths over time? Were sequences sampled entirely at random?

L133: Were sequences masked for problematic sites, i.e. due to homoplasy and likely sequencing errors, as identified by Nicola de Maio and group, prior to the phylogenetic inference? This is important and standard in these analyses; if the authors performed masking of problematic sites it is important to mention in the methods. If masking of problematic cites was not performed then this should be done and the analyses repeated during revision.

L156: why is the analysis delineated into periods in mid-July when the authors state here that the second European wave started in October? The delineation of epidemic waves and studied periods needs to be clarified and justified.

L161-165; L237-240; 374-376: the positive correlation between intra-territory transmission events and the estimated number of deaths does not convincingly validate the method, because this is what would be expected even in the presence of remaining bias. Why not total transmission in relation to deaths? Lastly, the authors have shown deaths are also associated with sequences, therefore this does not distinguish model accuracy from reflecting the relative inclusion of sequences.

L378: McLaughlin et al. 2022 used maximum likelihood ancestral reconstruction, but did not sum across replicates; rather, they averaged across replicates. Also, it would be good to include a couple other references as well that a similar approach, as well as the origins of the method algorithms (Pupko et al., 2000, Cunningham et al., 1998)

There are multiple instances in the Results section where relative flows are discussed for different geographies or over time, but only in a qualitative rather than qualitative manner. Please quantify rate estimates and/or relative changes/differences (as the authors did on L296 and L321) in other places. L186: 'at similar rates'. L192: 'quite similar'. L225: 'sequentially – and drastically – increased…'. L233: 'drastic'. L245: 'drastic'. L266: 'steep increase'. L268: 'sharp increase'. L294 'much more'. L308: 'a bit lower'.

L279: it is speculative that government measures had varying success rates based on the analyses conducted and presented in this manuscript.

L275; and generally: The manuscript would benefit from more discussion or stratification of which Pango lineages or Nextstrain clades dominated particular transmission types and routes in the two waves.

L299: Normally, Spearman rank correlation can be appropriately performed when there are 10 or more observations, here, as written the correlation has been performed with only three measurements – please clarify or rectify this.

L430; Supplementary: The GISAID identifiers should be a separate appendix and the authors need to follow the GISAID recommendations for inclusion of a table acknowledging labs for each sequence.

L443-446: What is missed by not conducting all three analyses together? How much international origin among a given region in France, for example?

L470: The inclusion of earlier diversity from the first wave in the trees for the second wave is laudable, however, how did the authors ignore transmission events involving those context sequences when analyzing the second wave in terms of intra- vs inter-territory transmission? Precisely how this was done needs to be clarified in the manuscript.

L480: How were trees rooted? This is an important omission and should be clearly specified in methods.

L483: how could the use of a strict clock rate have impacted the results? Employing model selection and describing in methods to justify would be valuable. If the authors have already performed model selection then clear description in the results is warranted.

L486: in regards to Figure 6-Figure supp 1 on estimating versus fixing the clock rate, the authors should move some of the text in the legend into the main results to better explain why the authors fixed a strict clock rate. Also, the reader may wonder to what extent the issue encountered with early node dating and long branches might have been related to specifying a strict instead of relaxed clock. Thus, please elaborate on this in the results.

Generally in the figure supplement legends, any interpretations of the plots should be moved into the Results section.

It would be informative to contextualize the authors' results within the literature available to date by having a more thorough comparison to other papers on the genomic epidemiology of SARS-CoV-2 in Europe in the Background and/or Discussion; for example, Worobey 2021, Hodcroft 2021, and Huisman et al. 2022.

---

## [Author Response]

Essential revisions:Both reviewers and the Editor think that this is an important study that presents interesting findings about the spread of SARS-CoV-2 in France and in Europe, and are positive about the work's eventual publication. However, the reviewers brought up important considerations, including methodological concerns, that need to be addressed in a revised version. Following the consultation with the editor and reviewers, we would welcome a revised version that addresses the substantial methodological comments. In summary:(1) Sampling: there are two questions. One is about the number of sequences and the number of deaths (both reviewers commented on this), as opposed to the number of infections; whether the correlation (sequences to deaths) is an artefact of the method used (Rev 1) and whether it validates the analysis (Rev2). In addition, since deaths and infections can be reported differently in different places, what is the impact of having different sampling fractions in different geographies? this could be addressed by looking at the robustness of the inferences to different choices of relative numbers of sequences from different regions

We thank the reviewers and editors for highlighting these various issues. First of all, the choice of sampling the number of sequences per week based on the number of deaths seems to us much more relevant than based on the number of infections. Indeed, confirming a death related to COVID-19 is globally robust and reported by health care centers, whatever the time period investigated. In sharp contrast, studying the number of infections completely biases any comparison between the first and second waves, since the availability of detection tools (RT-qPCR and antigenic tests) varied widely by country and throughout the year. PCR tests were almost not used outside of hospitals during the first wave in Europe, whereas their use has increased significantly in the second part of 2020. Therefore, we necessarily find many more infections than deaths during the second wave. Here we made this observation in France as an example (other countries were investigated and presented in the new Figure 1—figure supplement 2): for the first wave, about 30,000 deaths were recorded, and 34,000 during the second wave, suggesting that the virus had overall a similar impact on the French population, which we think is expected because no vaccine was available. Focusing on infections, there were about 215,000 confirmed infections in the first wave, and ten-fold more (about 2.5 million cases) during the second wave (based on COVID-19 data explorer from https://ourworldindata.org/). It is highly unlikely that the virus circulated ten-fold more in the population, since containment measures were still quite high as indicated by the stringency score index shown in Figure 1 (replacing the virus reproduction rate in the previous version). This difference cannot be explained by a change in virulence of the circulating strains, since the α variant started to increase in frequency only at the very end of 2020. Therefore, the drastic ratio of death to infection change is easily explained by the increase of testing during the year 2020. Since PCR tests were used in a large majority of countries from the summer of 2020 onwards, we thus believe that sampling the number of sequences on the basis of COVID-19-related deaths is a more accurate proxy than sampling from the number of confirmed infections. Since information is now stated at lines 165-171 and 458-464.

Concerning the correlation between the number of deaths and the number of sequences included for a subsample for a given time period of time and geographical scale, we aimed to show that our selection algorithm performs a consistent sampling with the epidemiological data, so as not to over- or under-represent a locality at a specific date. We however agree that the correlation of intra-regional transmission and the number of deaths is a very tautological argument (since our baseline sampling respects these spatio-temporal constraints), and is therefore not a direct validation of the method. We removed this element from the revised manuscript.

(2) Downsampling to < 1000 sequences, bootstrapping and adding: We were all concerned about the small individual datasets and the assumption that adding the results would generate the correct numbers of viral movements. After all, why this number of replicates – why not add after 5000 or 800 or some other number of replicates? Also, there are some instances where the same introduction event would be detected in many replicates, and somewhere it would be missed in most of them, so it's not clear that it's additive. Other authors have averaged, in similar analyses (but since each replicate might miss some events, averaging might not be). Given that this analytical method has the advantage that much larger numbers of sequences are tractable to analyze, the authors should analyze substantially larger datasets.

The low sampling and the production of some subsamples is, in our opinion, one of the strengths of our work and should be the subject of a precise justification. The use of about 1,000 sequences allows a non-negligible saving in computation time, both for phylogenetic reconstruction, temporal calibration, and the reconstruction of ancestral characters at nodes. To demonstrate the validity of multiple subsampling of the same pool of sequences, we included in the manuscript a simulation assay (lines 107-160 for the results, lines 475-502 for the method, figure 2 and figure 2—figure supplement 1). Based on our results, performing a number of subsamples greater than 100 is not useful. In our various analyses we systematically studied the number of introduction and exportation events per subsample, and calculated the mean and standard deviation over the course of the subsamples: we observed that each locality reaches an equilibrium of exchange events fairly quickly (often after 25 subsamples).

Summing exchange events by region and date across subsamples was unconventional. As recommended, we averaged the events by locality and week across all subsamples (the ratio between geographical units remains unchanged).

(3) The fast clock assumption has the potential to bias the analysis; a number of other related methodological issues were raised by reviewers. These should be addressed.

We would like to thank the editors and reviewers for this remark, as the molecular clock is a crucial feature of this phylodynamic study. Initially, our goal was primarily to correlate the number of SARS-CoV-2-related deaths with the number of intra-regional exchanges. To ensure that, we fixed a fast molecular clock, with a mutation rate about 10 times higher than expected (6 x 10^-3^ instead of ~7 x 10^-4^ substitution per site per year). After many tests, discussions and recommendations (in part with and by Dr. Angela McLaughlin, reviewer #2 of this work and first author of the genomic epidemiology of the first two waves of SARS-CoV-2 in Canada (doi: 10.7554/*eLife*.73896)), our initial attempt was not really justified and, as previously pointed out, the correlation between the number of intra-regional transmissions and the epidemiological data is a simple byproduct of the methodology. We agree that the choice of such a clock has the potential to strongly bias our observations. Therefore, we reassessed all our code and uncovered a simple error in our dating method. After correction of this error, the mutation rate – previously set at 6 x 10^-3^ substitution per site per year – is now estimated for each subsample and consistently with other studies, reaching ~7 x 10^-4^.

The other methodological concerns raised by the reviewers will be discussed specifically later.

(4) Quantitative descriptions of the results and their variability should be included in the results (for example, the importation rate). This will aid in articulating the public health interpretation and relevance, which reviewers also note would add interest and weight to the paper.

In the first version of our study, we mainly presented the results in a qualitative way. We agree with the editors and reviewers that a quantitative aspect is required for having a much more precise view of the exchanges between the different localities, but also of the main changes between the first and second epidemic waves. We therefore reworked all the results to put more emphasis on the quantitative aspect of the exchanges between localities. Now, we systematically present the percentages of introduction and exportation events in order to have a more precise idea of which continents/European countries/French regions exchanged the most.

Reviewer #1 (Recommendations for the authors):The method is described as 'original' but line 377 implies that the same method was used to study data from Canada. Please clarify what is new about the approach used here relative to previous studies.

The similarity with a pre-existing study was a clear overstatement on our part. McLaughlin *et al.* 2022 (*eLife*, doi: 10.7554/*eLife*.73896) studied the genomic epidemiology of the first two waves of SARS-CoV-2 in Canada without resorting to Bayesian inference. The authors also used a subsampling strategy, however, the aim was to produce subsets using different proportions of Canadian sequences to identify the proportion maximizing the ability to identify domestically circulating sublineages (they ended up with 27,552 sequences, 75% of their total). The re-sampling strategy is indeed ‘original’ and we added a simulation assay to show its performance. All this information has been clarified in the revised version.

It is stated in line 65 that existing phylogenetic tools cannot include the large datasets generated by projects such as the international COVID sequencing effort. However, researchers on the COG-UK project generated a tree of sequences comprising hundreds of thousands, if not millions of sequences (https://www.cogconsortium.uk/priority-areas/data-linkage-analysis/public-data-analysis/). It would be helpful to clarify what can and cannot be achieved with which phylogenetic methods.

Building a phylogeny of several hundreds of thousands or millions of sequences is theoretically conceivable. When we look in depth at the astounding work done by COG-UK, we can see on the methods described on their ‘phylopipe’ github (https://github.com/virus-evolution/phylopipe) that they do not build a gigantic tree in one go but instead use a smart divide and conquer strategy bases on computing subtrees of known clades that they graft together at the end.

Whether using Bayesian or maximum likelihood approaches, dating such phylogenies and reconstructing ancestral characters at nodes (for example, the Pango lineages constituted of hundreds of factors) can require several weeks – if not more – of computational time. Therefore, we chose a strategy to overcome these limits by considering relatively small phylogenies across different sampling (i.e., subsamples). This information has been clarified in the revised version of the manuscript, especially at lines 357-360.

The positive correlations observed between the number of deaths and the number of sequences in a dataset (Figure 2) is a reassuring check but seems to be a direct artefact of the methods, whereby the number of deaths was used to determine the number of sequences. I suggest including these plots in supplemental material.

We agree that the positive correlations between the number of SARS-CoV-2-related deaths and the number of sequences in a set are simply a result of a sequence draw consistent with the epidemiological data. The plots were made to show that the datasets we produced do not include notable biases. As advised, these plots were put as a supplementary figure (figure 2—figure supplement 2), while we put more emphasis on a simulation analysis to prove that a set of 1,000 sequences gave satisfying results in describing more complex phylogenies.

I would value more explicit information in the Methods section about how data from the inferences performed on different bootstrapped samples was combined.

In the previous version, exportation and introduction events were summed, which is indeed not the most conventional strategy. In the revised version, we averaged the events within subsamples, by locality and by week. It did not change any trend or ratio but it is more standard now. This clarification has been introduced in the “Methods” section (lines 536-537).

The link to the stated GitHub repository did not work at the time I tried it.

The link to access the scripts produced for this study did not work. This has been corrected, and anyone can use and modify the scripts for their own works/projects.

Reviewer #2 (Recommendations for the authors):L97: Figure comments: It is recommended to add an annotation for the date delineation of the two COVID waves analyzed in the main graphic. The legend should report how reproduction rate was calculated, and this should be clearly described in the methods. Relatedly, L447-448: state exact dates instead of 'mid' and 'late' months. Applies in results as well.

We annotated in the Figure 1—figure supplement 2 the two time periods investigated in the study, as well as in some parts in the main text. As far as the reproduction rate is concerned, since we did not talk about it or relate it to our study, it has been removed from the figure. Instead, we put the stringency index, which seems to us much more relevant. The definition of this index is explained in the legend of Figure 1 (lines 695-698). For lines 447-448 (as well as in other parts in the manuscript, notably the “Results” section), we modified the text by giving the exact dates rather than the notions of “mid” or “late”.

L106: Although the authors say that sequences are representative of 'sars-cov-2 infections', they then describe the comparison to deaths. Did the authors also compare case incidence to deaths and separately, to sequences over time? Need to better justify the use of deaths rather than new diagnoses. It would help the reader if there was also a presentation of cases over time in addition to deaths that could be done for example in Figure 1.

This is a misnomer on our part. The subsamples we produced are spatio-temporally representative of COVID-19-related deaths, not infections. We did not produce subsamples based on infections because we believe that this parameter is biased due to the heavy use of PCR tests only from the second epidemic wave, wrongly suggesting that the virus circulated much more during the second wave than the first wave. For more details, we refer the reviewer to point (1) of the essential revisions.

L114;L124;L209;L287,L465: Regarding the discarding of sequences from geographies that were underrepresented in the sequence set – how would the results be expected to change if these geographies were not discarded? They would be perhaps less likely to be sampled as well as represented in the ancestral nodes due to low representation, but they might have indicated additional nuances to the French regional transmission patterns.

The reviewer's remark is very pertinent, and we asked ourselves this question at the beginning of this study: should we or not discard countries/regions for which the number of sequences was (significantly) underestimated? We decided to discard such countries/regions in order to avoid misinterpretation. If we take the example of the French regions, based on the few reports from Santé Publique France (https://www.santepubliquefrance.fr/dossiers/coronavirus-covid-19), the Grand-Est region would have been strongly involved in the dissemination of the virus during the first French wave, in particular because of the evangelical gathering in Mulhouse, in the Haut-Rhin, where more than a thousand people were contaminated. This was one of the largest epidemic clusters in Europe at the beginning of the epidemic. However, on the GISAID platform, only a handful of sequences associated with the Haut-Rhin region were available, and we could not have reproduced such an observation. In the new version, we however added two additional French regions, Bretagne and Occitanie, which include some others very densely populated French cities (Toulouse, Montpellier and Rennes).

Also, in the methods, what was the threshold by which the authors decided if a region was too underrepresented to be included? L404: what percent of cases and/or deaths were associated with the administrative regions that were excluded due to low or no sequences in each wave?

This information was missing from the “Methods” section. We corrected that in the new version of the manuscript. Briefly put as can be seen in Figure 2—Figure Supplement 3, the regions were very unevenly represented in the data. We excluded regions with more than 20% of weeks insufficiently represented in the database (lines 464-469).

L115: Need to clarify how many sequences per phylogeny in the results. Were sequences sampled by geography randomly or proportionally to the relative number of deaths over time? Were the sampled sequences distributed uniformly over time or proportionally to deaths over time? Were sequences sampled entirely at random?

Before producing a phylogeny, we perform a random draw of sequences per country/French region and per week. For that, we first specify the desired sample size (e.g., 1,000 sequences). The next step consists in assigning the number of sequences associated with each country/French region according to the number of deaths related to COVID-19, and then to make a breakdown by week, still according to the number of deaths. In the revised version of the manuscript, we detail this procedure in the “Results” and "Methods" section. The dataset generation algorithm is provided on GitHub (https://github.com/Rcoppee/PhyloCoV). The figure showing the correlation, for a subsample, between the number of sequences included per continent/European country/French region and per week as a function of the number of deaths represents, according to us, a proof of the well constitution of the subsamples.

L133: Were sequences masked for problematic sites, i.e. due to homoplasy and likely sequencing errors, as identified by Nicola de Maio and group, prior to the phylogenetic inference? This is important and standard in these analyses; if the authors performed masking of problematic sites it is important to mention in the methods. If masking of problematic cites was not performed then this should be done and the analyses repeated during revision.

We mask the problematic sites using the list provided by De Maio and colleagues (filedate=2021-10-27) on GitHub. All the sites listed in the vcf file were changed to N’s after the alignment step. This has been now precised in the text (lines 506-509).

L156: why is the analysis delineated into periods in mid-July when the authors state here that the second European wave started in October? The delineation of epidemic waves and studied periods needs to be clarified and justified.

The second European epidemic wave effectively begins in October with an almost exponential increase in the number of SARS-CoV-2-related deaths. However, on a continental scale, Asia and South America (and to a lesser extent, North America) were particularly affected by the pandemic during the summer of 2020 (between June and September, based on COVID-19 data explorer from https://ourworldindata.org). Therefore, we thought it would be interesting to study the role of Europe during this period. Furthermore, while the epidemic has strongly declined in Europe in summer 2020, some European countries such as Russia continued to have a relatively high number of COVID-19-related deaths. It was therefore interesting for us to better understand what this epidemic in Russia may have entailed. In our study, it appears that Russia likely transmitted the virus to a number of European countries, possibly causing a resurgence of cases in Europe, particularly in Spain, before proliferating to the rest of Europe.

We clarified the concepts of time periods by giving the precise dates of each period to avoid ambiguity thorough the revised version of the manuscript.

L161-165; L237-240; 374-376: the positive correlation between intra-territory transmission events and the estimated number of deaths does not convincingly validate the method, because this is what would be expected even in the presence of remaining bias. Why not total transmission in relation to deaths? Lastly, the authors have shown deaths are also associated with sequences, therefore this does not distinguish model accuracy from reflecting the relative inclusion of sequences.

The reviewer made broadly the same point as Reviewer 1. We agree with their observations: the positive correlation between the number of intra-regional events and the number of deaths related to COVID-19 is only an artifact of the methodology. In the revised version of the manuscript, we removed all such notions of correlations because they did not ultimately add any weight to the observations. The number of intra-region transmissions was simply used to determine the total proportion of inter-region transmissions, and thus quantify the increase or decrease in the number of such events between the two time periods investigated.

L378: McLaughlin et al. 2022 used maximum likelihood ancestral reconstruction, but did not sum across replicates; rather, they averaged across replicates. Also, it would be good to include a couple other references as well that a similar approach, as well as the origins of the method algorithms (Pupko et al., 2000, Cunningham et al., 1998)

As in the work of McLaughlin et al. – which serves as a benchmark for our present work – we averaged the observations across subsamples in the revised version. We also included the two proposed citations introducing a similar “replicate” approach (line 373).

There are multiple instances in the Results section where relative flows are discussed for different geographies or over time, but only in a qualitative rather than qualitative manner. Please quantify rate estimates and/or relative changes/differences (as the authors did on L296 and L321) in other places. L186: 'at similar rates'. L192: 'quite similar'. L225: 'sequentially – and drastically – increased…'. L233: 'drastic'. L245: 'drastic'. L266: 'steep increase'. L268: 'sharp increase'. L294 'much more'. L308: 'a bit lower'.

We thank the reviewer for this comment because we feel that including many more quantitative, rather than qualitative, results give more weight to our observations. In addition to the methodology, we rewrote much of the “Results” section to provide a multitude of numerical data, providing a much better understanding of the overall involvement of each geographic region in COVID-19 transmission.

L279: it is speculative that government measures had varying success rates based on the analyses conducted and presented in this manuscript.

This concluding sentence of the results was indeed very presumptuous on our part and has been removed in the revised version.

L275; and generally: The manuscript would benefit from more discussion or stratification of which Pango lineages or Nextstrain clades dominated particular transmission types and routes in the two waves.

The reviewer rightly asked if it was possible to see which lineages dominated the exchanges at the different geographic scales. In the revised manuscript, we performed a reconstruction of ancestral states at nodes including both geographic region and Pango lineage. In this way, we were able to study, notably in relation to France, which lineages were strongly introduced in this country and, conversely, which were the main lineages exported from France. These analyses were carried out at the different periods of time investigated. Novel items were added in the main figures to summarize these results.

L299: Normally, Spearman rank correlation can be appropriately performed when there are 10 or more observations, here, as written the correlation has been performed with only three measurements – please clarify or rectify this.

These correlations have been removed in the revised version, as previously explained.

L430; Supplementary: The GISAID identifiers should be a separate appendix and the authors need to follow the GISAID recommendations for inclusion of a table acknowledging labs for each sequence.

In the revised version, we followed the recommendations provided by the GISAID application. A specific page presenting all the sequences included in the analysis is dedicated, with all the participating authors and laboratories (lines 432-435).

L443-446: What is missed by not conducting all three analyses together? How much international origin among a given region in France, for example?

Conducting an analysis by aggregating the different geographical scales into a single dataset is not compatible with our strategy of producing small phylogenies. Indeed, on the continental scale and in the case of small phylogenies, we only included few sequences per week and per country (depending on the number of deaths). Therefore, we could not model with sufficient accuracy the exchanges for smaller geographical scales such as the European scale or the scale of French regions. The separation of geographical scales and time periods allows us to include sufficiently “large” sets of sequences with a relatively limited computational time cost.

L470: The inclusion of earlier diversity from the first wave in the trees for the second wave is laudable, however, how did the authors ignore transmission events involving those context sequences when analyzing the second wave in terms of intra- vs inter-territory transmission? Precisely how this was done needs to be clarified in the manuscript.

After further testing, we found that the presence of some sequences from the first time period was not essential to produce phylogenies with proper dating and tMRCA. Therefore, we no longer included sequences from the first time period in the second time period set.

L480: How were trees rooted? This is an important omission and should be clearly specified in methods.

It was indeed an unforgivable omission and we amended our method in the manuscript (lines 515-517). We simply define Wuhan-Hu-1/NC_045512 as an outgroup in LSD2 v2.3 for rooting.

L483: how could the use of a strict clock rate have impacted the results? Employing model selection and describing in methods to justify would be valuable. If the authors have already performed model selection then clear description in the results is warranted.

The issue of molecular clock was, in our opinion, the main bias of our work and was used to correct discrepancies in dating that were due to a bug in our code. We would like to thank once again reviewer #2 with whom we were able to exchange and who gave us many advices on the good use of the clock and to eliminate the possible biases. In this revised version, and after code proofreading, the molecular clock is no longer fixed but estimated using LSD2 v2.3 following the production of phylogenies with FastTree. The mutation rate for each replicate was very stable across the subsamples, around 7 x 10^-4^ substitution per site per year, a value close to those found in other studies (Xia *et al.* 2021, Viruses, doi: 10.3390/v13091790). Concerning the dating of events, we added some clarifications, and in particular on the fact that we considered the middle of the branch as the date of the event (lines 534-535).

L486: in regards to Figure 6-Figure supp 1 on estimating versus fixing the clock rate, the authors should move some of the text in the legend into the main results to better explain why the authors fixed a strict clock rate. Also, the reader may wonder to what extent the issue encountered with early node dating and long branches might have been related to specifying a strict instead of relaxed clock. Thus, please elaborate on this in the results.

Following correction of our molecular clock, this additional figure has been removed.

Generally in the figure supplement legends, any interpretations of the plots should be moved into the Results section.

As recommended by the reviewer, we tried to limit the interpretations and presentations of results in the figure legends.

It would be informative to contextualize the authors' results within the literature available to date by having a more thorough comparison to other papers on the genomic epidemiology of SARS-CoV-2 in Europe in the Background and/or Discussion; for example, Worobey 2021, Hodcroft 2021, and Huisman et al. 2022.

We included comparisons with other studies in the “Discussion” section, highlighting convergences and divergences about genomic epidemiology in Europe in 2020.